# Higher Concentration of Dietary Selenium, Zinc, and Copper Complex Reduces Heat Stress-Associated Oxidative Stress and Metabolic Alteration in the Blood of Holstein and Jersey Steers

**DOI:** 10.3390/ani12223104

**Published:** 2022-11-10

**Authors:** A-Rang Son, Seon-Ho Kim, Mahfuzul Islam, Song-Jin Oh, Man-Jeong Paik, Sung-Sill Lee, Sang-Suk Lee

**Affiliations:** 1Ruminant Nutrition and Anaerobe Laboratory, Department of Animal Science and Technology, Sunchon National University, Suncheon 57922, Korea; 2Department of Microbiology and Parasitology, Sher-e-Bangla Agricultural University, Dhaka 1207, Bangladesh; 3College of Pharmacy, Sunchon National University, Suncheon 57922, Korea; 4Institute of Agriculture and Life Science and University-Centered Laboratory, Gyeongsang National University, Jinju 52828, Korea

**Keywords:** antioxidant enzyme, dietary minerals, heat shock protein, heat stress, ruminants, serum metabolites

## Abstract

**Simple Summary:**

Heat stress has a negative effect on feed intake, growth, and metabolic disorders of ruminants that cause oxidative stress by the overproduction of free radicals and disturbance of the redox balance. Different breeds have different abilities to adapt to heat stress; thus, feeding them with antioxidants such as minerals can help build up an antioxidant defense system that prevents oxidative damage in the body. This study investigated the effects of recommended and high mineral concentrations on antioxidant enzymes, heat shock proteins, and metabolites in the blood of Holstein and Jersey steers during heat stress. Our results showed that animals supplemented with high mineral concentrations prevent oxidative stress and metabolite changes related to heat stress in the blood. Therefore, increasing mineral supplementation during high temperatures is an effective strategy for reducing heat stress.

**Abstract:**

This study investigated the influence of high concentrations of dietary minerals on reducing heat stress (HS)-associated oxidative stress and metabolic alterations in the blood of Holstein and Jersey steers. Holstein steers and Jersey steers were separately maintained under a 3 × 3 Latin square design during the summer conditions. For each trial, the treatments included Control (Con; fed basal TMR without additional mineral supplementation), NM (NRC recommended mineral supplementation group; [basal TMR + (Se 0.1 ppm + Zn 30 ppm + Cu 10 ppm) as DM basis]), and HM (higher than NRC recommended mineral supplementation group; [basal TMR + (Se 3.5 ppm + Zn 350 ppm + Cu 28 ppm) as DM basis]). Blood samples were collected at the end of each 20-day feeding trial. In both breeds, a higher superoxide dismutase concentration (U/mL) along with lower HSP27 (μg/L) and HSP70 (μg/L) concentrations were observed in both mineral-supplemented groups compared to the Con group (*p* < 0.05). The HM group had significantly higher lactic acid levels in Jersey steers (*p* < 0.05), and tended to have higher alanine levels in Holstein steers (*p* = 0.051). Based on star pattern recognition analysis, the levels of succinic acid, malic acid, γ-linolenic acid, 13-methyltetradecanoic acid, and tyrosine decreased, whereas palmitoleic acid increased with increasing mineral concentrations in both breeds. Different treatment groups of both breeds were separated according to the VIP scores of the top 15 metabolites through PLS–DA analysis; however, their metabolic trend was mostly associated with the glucose homeostasis. Overall, the results suggested that supplementation with a higher-than-recommended concentration of dietary minerals rich in organic Se, as was the case in the HM group, would help to prevent HS-associated oxidative stress and metabolic alterations in Holstein and Jersey steers.

## 1. Introduction

Heat stress (HS) adversely affects ruminant production, which is considered an increasing threat to the livestock industry and causes great economic losses globally [1,2,3,4]. Furthermore, the third assessment report of the International Panel on Climate Change predicted that environmental temperatures would rise by 1.4–5.8 °C everywhere on Earth from 1990 to 2100 [5]. Indeed, all seasons in the Korean Peninsula experienced an increase of 2 °C from 1992 to 2004. In addition, the number of summer days in Korea increased from 1908 to 2009 [6]. The decrease in dry matter intake (DMI) of cattle is the most common deleterious effect of HS, as it affects energy and protein metabolism, increases metabolic disorder and mineral imbalance, and disturbs the secretion of hormones and blood metabolites, among other health problems [7,8,9]. For instance, Holstein steers showed a significant reduction in DMI while undergoing HS [10]. Furthermore, metabolites in ruminant biofluids can be affected by the species of ruminant, type of feed consumed, breeding environment and season [11]. HS alters metabolic pathways in cattle, such as upregulating glycolysis and the tricarboxylic acid cycle (TCA) [12,13]. In addition, reactive oxygen species (ROS) production exceeds the concentration of antioxidant enzymes present in the body during HS, leading to oxidative stress in cattle [14,15,16,17]. Increased ROS production induces heat shock protein (HSP) expression [18]. Superoxide dismutase (SOD), an antioxidant enzyme, helps neutralize superoxides produced in the body [19]. Therefore, dietary supplementation with antioxidants is required during HS to balance ROS and antioxidants and prevent oxidative stress.

Several essential trace minerals, such as zinc (Zn), copper (Cu), and selenium (Se), have antioxidative properties and are used as dietary supplements in ruminants [20,21,22]. Both Zn^2+^ and Cu^2+^ are able to act as co-factors of SOD for enzymatic activity [23,24], while Se can act as a scavenging antioxidant [25,26]. The organic form of Se (selenomethionine and yeast Se) performs better than inorganic Se (sodium selenite or selenate) [22]. Likewise, organic forms of Zn and Cu have shown more beneficial effects in ruminants [21]. Weng et al. [27] observed the beneficial effects of Zn supplementation at a maximum dose of 75 ppm/kg DM in lactating dairy cows during HS, while the dosage of Cu ranged from 10 to 30 ppm for beef cattle [28]. Sun et al. [29] observed that 0.3 ppm/kg DM of organic selenium was beneficial in reducing oxidative stress in cattle. However, a definitive optimum dose of these minerals has not been set; the dose recommended by the National Research Council (NRC) [30] is not sufficient to successfully overcome the adverse effects of HS during summer, and this is a challenging issue in the dairy industry. We hypothesized that higher concentrations of dietary minerals have antioxidant potential; however, they do not exceed the maximum tolerable level recommended by the NRC, and can effectively and sustainably reduce the adverse effects of HS in ruminants. Moreover, the sensitivity and response to HS vary among breeds. Previous studies have shown that both Jersey steers and dairy cows are less sensitive to HS than Holstein steers [10,31]. Therefore, both Holstein and Jersey steers were included in the present study. Considering the aforementioned facts, the present study was conducted to evaluate the influence of higher concentrations of dietary mineral (Zn, Cu, and Se) mixtures on the serum biochemistry, enzyme activity, HSP, and metabolites of Holstein and Jersey steers during HS.

## 2. Materials and Methods

### 2.1. Animals, Experimental Design, and Diet

Three non-cannulated Holstein (710.33 ± 43.02 kg; ~30 mo) and three Jersey (559.67 ± 32.72 kg; ~32 mo) steers were maintained under the two separate 3 × 3 Latin square design. For both trials, treatments included a Control group (without additional mineral supplementation and only comprising basal TMR), an NM-supplemented group [TMR + NRC recommended concentration of mineral supplementation (Se 0.1 ppm + Zn 30 ppm + Cu 10 ppm) as DM basis], and a HM-supplemented group [(basal TMR + (Se 3.5 ppm + Zn 350 ppm + Cu 28 ppm) as DM basis]. Organic forms of Se (Yeast-Selenium; X-SEL 3000TM, Algebra Bio, New South Wales 2041, Australia), Zn-glycenate (BASF SE, Ludwigshafen 67056, Germany), and Cu-glycenate (BASF SE, Ludwigshafen 67056, Germany) were used to supplement Se, Zn, and Cu, respectively. A feeding trail of 20 days was conducted in each period, and blood samples were collected on the last day. A 7-day washing period was maintained between each Latin square to return the mineral concentration as baseline level [32,33,34]. The ingredients and chemical compositions of the basal TMR are presented in Table 1. All steers were maintained in individual stalls with feeding and water facilities. The Con group of steers was fed only basal TMR once a day at 0900 h at a rate of 5–10% of refusal, while the respective concentrations of minerals were mixed well with the basal TMR for NM and HM groups. The DMI was measured as the difference between the feed offered and feed refusal. The basal TMR was sampled twice (on days 7 and 14) during the feeding trial and the dry matter content was determined using a hot-air oven at 65 °C for 72 h [35]. The chemical composition of the TMR was analyzed using standard methods [36]. The neutral detergent fiber (NDF) and acid detergent fiber (ADF) contents were determined according to protocols described by Van Soest et al. [37] and Van Soest [38], respectively. The average daily gain (ADG) was calculated as (final body weight − initial body weight)/number of experimental days in each period.

### 2.2. Recording of Temperature-Humidity Index (THI)

This study was conducted under moderate heat stress conditions with an average THI value of 82.79 ± 1.10 (Table 2). The ambient temperature (°C) and relative humidity (%) of the experimental shed were recorded throughout the experimental periods using the Testo 174H Mini data logger (West Chester, PA, USA). The THI was calculated as THI = (0.8 × ambient temperature) + [% relative humidity/100 × (ambient temperature – 14.4)] + 46.4 [39].

### 2.3. Sample Collection, and Processing

On the last day of the feeding trial in each period, 15 mL blood samples were collected from the jugular vein of steers before the morning meal and immediately, 10 mL was transferred into two serum separator tubes (SSTTM II advance, BD vacutainer^®^) and 5 mL into a different serum tube (Trace Element serum, BD vacutainer^®^). After centrifugation at 4000 rpm for 10 min, serum samples were transferred into 2 mL micro tubes and stored at −80 °C to analyze serum chemistry, SOD, HSP concentration, serum trace minerals, and serum metabolites.

### 2.4. Analysis of Serum Biochemistry, Trace Minerals, SOD, and HSPs

Serum calcium (Ca), magnesium (Mg), phosphorus (P), BUN, TP, and CHOL levels were analyzed using a biochemical analyzer (IDEXX Catalyst OneTM, IDEXX Laboratories, Inc., Westbrook, ME, USA). Serum glucose was analyzed using a portable glucose test meter (FreeStyle Optium Neo H Blood Glucose and Ketone System; Abbot Diabetes Care Ltd., Maidenhead, UK). The serum concentrations of Zn, Cu, and Se were analyzed using an inductively coupled plasma mass spectrometry (ICP) system (ELAN DRCe, PerkinElmer Waltham, MA, USA). Serum SOD concentration was measured using the Superoxide Dismutase Assay Kit (Item No.706002, Cayman Chemical Company, Ann Arbor, MI, USA) following the manufacturer’s instructions. Serum HSP 27 and HSP 70 concentrations were measured using the Bovine Heat Shock Protein 27 (HSP27) ELISA Kit and Bovine Heat shock-related 70 kDa protein 2 ELISA Kit, respectively (MyBioSource, Inc., San Diego, CA, USA) according to the manufacturer’s instructions.

### 2.5. Analysis of Serum Metabolites

#### 2.5.1. Gas Chromatography–Tandem Mass Spectrometry

Serum metabolomic profiling analysis of organic acids (OAs) and fatty acids (FAs) was performed by using gas chromatography–tandem mass spectrometry (GC–MS/MS) using the multiple reaction monitoring (MRM) mode, as described previously [40]. The simultaneous analysis was performed by using a Shimadzu 2010 Plus gas chromatograph equipped with a Shimadzu TQ 8040 triple quadruple mass spectrometer (Shimadzu, Kyoto, Japan) coupled with an Ultra-2 (5% phenyl–95% methylpolysiloxane bonded phase; 25 m × 0.20 mm i.d., 0.11 μm film thickness) cross-linked capillary column (Agilent Technologies, Atlanta, GA, USA). The samples were injected in the split mode (10:1). The GC oven temperature was initially set at 100 °C for 2 min and increased to 300 °C at a rate of 10 °C/min with an 8-minute holding time. The temperatures of the injector, interface, and ion source were 260, 300, and 230 °C, respectively. The carrier gas was helium at a flow rate of 0.66 mL/min and argon was used as collision gas. Electron impact mode at 70 eV was performed as the ionization mode. The collision energy (CE) varied from 3 to 45 V in increments of 3 V in MRM mode.

#### 2.5.2. Sample Preparation for Profiling Analysis of OAs and FAs in Sera

Profiling of OAs and FAs was conducted with methoxime/tert-butyldimethylsilyl (MO/TBDMS) derivatives, as was previously described [41,42,43]. Briefly, proteins were removed using acetonitrile (150 μL) to 50 μL of serum containing 0.1 μg of 3,4-dimethoxybenzoic acid and lauric-d_2_-acid as internal standards (ISs). After centrifugation, the supernatant was added to 800 μL distilled water. The aliquot solutions were adjusted to pH ≥ 12 with 5.0 M sodium hydroxide and MO derivative was conducted by using methoxyamine hydrochloride at 60 °C for 60 min. The aqueous phase as sequential MO derivative was adjusted (pH ≤ 2.0) with 10% sulfuric acid, saturated with sodium chloride, and extracted with ethyl acetate (2 mL), a mixture of ethyl acetate (2 mL), and diethyl ether (3 mL). The extracts were evaporated to dryness under a gentle nitrogen stream. To form TBDMS derivative, the dry residues containing OAs and FAs were reacted at 60 °C for 60 min with triethylamine (TEA) (5 μL), toluene (10 μL), and MTBSTFA (20 μL). For quantification analysis, standard samples including ISs, OAs, and FAs at different concentrations (0.01–5.0 μg/mL) were subjected to sequential MO/TBDMS derivatives to obtain a calibration curve as described above. All samples were individually prepared in triplicate and analyzed by GC–MS/MS in MRM mode.

#### 2.5.3. Liquid Chromatography–Tandem Mass Spectrometry

Serum samples (20 μL), ^13^C_1_ phenylalanine as IS (50 ng), and ACN (60 μL) were added and centrifuged for 3 min at 12,300× *g*. Supernatants were filtered using Spin-X centrifuge filters [0.22 μm, Costar, Corning Incorporated, Corning, NY, USA)]. Samples (1 μL) were then injected into the LC–MS/MS system (LCMS-8050, Shimadzu Corp., Kyoto, Japan) using an autosampler. This method was used to profile the 49 amino acids. For quantification analysis, standard samples including IS and AAs at different concentrations (0.005–2.0 μg/mL) were subjected to obtain calibration curve as described previously. Chromatographic separations were performed using Intrada amino acid (50 mm × 3.0 mm, 3 μm) at a flow rate of 0.6 mL/min using mobile phases A [ACN/THF/25 mM ammonium formate/acetic acid = 9/75/16/0.3 (*v*/*v*/*v*/*v*]) and B (ACN/100 mM ammonium formate = 20/80 (*v*/*v*)]. The applied gradient was as follows: 0% B for 2.5 min, increased to 17% B (6.5 min) and to 100% B (10 min) and decreased to the initial 0% B (12 min), followed by a 3 min re-equilibration period. MS/MS was performed in electrospray ionization (ESI) mode. Column oven, autosampler, interface, desolvation line, and heat block temperatures were 40, 4, 200, 200, and 300 °C, respectively, and flow rates of nebulizing gas, drying gas, and heating gases were 3.0, 10.0, and 10.0 L/min, respectively. The pressure of the collision-inducing dissociation gas was set at 270 kPa.

#### 2.5.4. Star Pattern Recognition and Multivariate Statistical Analysis in Sera

The levels of OAs, AAs, and FAs in serum samples from different treatment groups were determined on the basis of their calibration curves, which were expressed as percentage composition (%). The composition values were normalized versus the Con group. Star pattern recognition analysis was performed with normalized values using Microsoft Excel (Microsoft, Redmond, WA, USA) [44]. Kruskal–Wallis test was used to observe significant differences among the groups of metabolites. Multivariate analysis was performed using principal component analysis (PCA) and partial least-squares discriminant analysis (PLS–DA) on MetaboAnalyst (https://www.metaboanalyst.ca (accessed on 4 March 2021)). The PLS–DA model was validated by the correlation coefficient (R2) and cross-validation correlation coefficient (Q2) using statistical parameters [44,45,46,47,48].

### 2.6. Statistical Analysis

All data on growth performance, serum biochemistry, trace minerals, SOD, and HSPs of each breed were analyzed separately using the general linear model along with Duncan’s multiple range test using SAS (version 9.4; SAS Institute Inc., Cary, NC, USA) [49]. The model included the fixed effect of treatment, and the random effects of animal and period where each animal was distributed under 3 different treatment groups in 3 different periods. Additionally, the growth performance, serum biochemistry, and trace minerals data were analyzed using the mixed procedure of SAS where the breed and treatment (trace mineral supplementation) were considered as factors. Statistical significance was set at *p* < 0.05.

## 3. Results

### 3.1. Growth Performance, Serum Biochemistry, Trace Minerals, SOD, and HSPs

The results of growth performance, serum biochemistry and trace mineral concentrations of Holstein and Jersey steers supplemented with different minerals are presented in Table 3. The DMI (kg/d) was not influenced either by breed or the trace mineral supplementation (*p* > 0.05); however, Jersey steers had higher AGD (kg) and FE compared to Holstein steers (*p <* 0.05). All the tested serum biochemical parameters, such as Ca, Mg, P, blood urea nitrogen (BUN), glucose, total protein (TP), and cholesterol (CHOL), remained unchanged among the different treatment groups in both breeds (*p* > 0.05). In Holstein steers, the highest serum Se concentration was observed in the HM, followed by NM, and Con groups (*p <* 0.05). Similarly, in Jersey steers, the HM and NM groups had a significantly higher concentration of serum Se compared to the Con group (*p <* 0.05). In contrast, serum concentrations of Cu and Zn were observed to be similar among the different treatment groups in both breeds (*p* > 0.05). In addition, according to the mixed analysis, a numerical increase in the serum Cu concentration (*p >* 0.05) along with a significant increase (*p <* 0.05) in the concentrations of serum Zn and Se were observed along with the supplementation of trace minerals, regardless of breeds. However, Jersey steers had a significantly higher serum Zn concentration compared to Holstein steers regardless of treatment (*p <* 0.05).

The results of additional mineral supplementation on serum SOD and HSPs in both breeds are presented in Figure 1 and Figure 2. Dietary supplementation with a higher concentration of trace minerals had a significant influence on serum SOD and HSPs in both breeds. In the case of Holstein steers, both mineral-supplemented groups (NM and HM) had a significantly higher concentration of SOD (U/mL) than the Con group (Figure 1A) (*p* < 0.05). However, in Jersey steers, the HM group had the highest SOD concentration (U/mL), followed by the NM and Con groups (Figure 1B) (*p* < 0.05). In contrast, the HM group had a significantly lower concentration of HSP27KDa (ng/mL) and HSP70KDa (ng/mL) than the Con group in Holstein steers (*p* < 0.05) (Figure 2A,C). Similarly, in Jersey steers, a lower concentration of HSP27KDa (ng/mL) was observed in the HM group than in the other groups, while both the HM and NM groups had a lower concentration of HSP70KDa (ng/mL) than the Con group (*p* < 0.05) (Figure 2B,D).

### 3.2. Serum Metabolites

For both Holstein and Jersey steers, a total of 54 metabolites, including 12 OAs, 16 FAs, and 26 AAs, were determined by GC–MS/MS and LC–MS/MS in MRM mode and presented as their percentage composition (%) in Appendix A, respectively. Among the 12 OAs, lactic acid (No. 2) was the most abundant, followed by glycolic acid (No. 3), 3-hydroxybutyric acid (No. 6), and hippuric acid (No. 12), in all treatment groups for both breeds. Among the 16 FAs, docosatetraenoic acid (No. 24) was the most abundant, followed by stearic acid (No. 20) in all treatment groups for both breeds. Oleic acid (No. 18), linoleic acid (No. 17), arachidonic acid (No. 21), and palmitic acid (No. 15) were the other abundant FAs (>10% abundance) in all treatment groups in both breeds. Among the 26 AAs, glutamine (No. 45) was the most abundant AA, followed by creatine (No. 44) and valine (No. 35), in all treatment groups for both breeds. Leucine (No. 32) and alanine (No. 42) were also abundant AAs, with an abundance of > 5% in all treatment groups in both breeds. According to the Kruskal–Wallis test, none of the metabolites differed significantly among the treatment groups in either breed (*p* > 0.05), except for lactic acid, which was significantly higher in the HM group than in the other Jersey steers (*p* < 0.05). Alanine tentatively increased with increased mineral supplementation in Holstein steers (*p* = 0.051).

The star pattern recognition analysis of 12 OAs, 16 FAs, and 26 AAs is shown in Figure 3. Of the 12 OAs, the levels of succinic acid (No. 7) and malic acid (No. 9) decreased with increasing mineral concentration in both breeds. In contrast, an increased pattern of pyruvic acid (No. 1) was observed with increased mineral supplementation in Jersey steers. A lower level of hippuric acid (No. 12) was observed in the HM group than in the other groups in both breeds. In terms of FAs, the levels of γ-linolenic acid (γ-C_18:3_) (No. 16) and 13-methyltetradecanoic acid (C_14:0_) (No. 26) decreased, whereas palmitoleic acid (C_16:1_) (No. 14) increased with increasing mineral concentrations in both breeds. In addition, the levels of linoleic acid (C_18:2_) (No. 17) and α-linolenic acid (α-C_18:3_) (No. 19) were lower in the HM and NM groups than in the Con group for both breeds. In addition, the level of eicosadienoic acid (C_20:2_) (No. 22) decreased with higher mineral supplementation in Jersey steers. Furthermore, increased levels of myristic acid (C_14:0_) (No. 13) and oleic acid (C_18:1_) (No. 18) in Holstein along with increased levels of palmitic acid (C_16:0_) (No. 15) and docosapentaenoic acid (C_22:5_) (No. 25) in Jersey steers were observed with higher concentrations of mineral supplementations. In terms of AAs, the level of phenylalanine (No. 30) increased, whereas that of tyrosine (No. 31) decreased with increasing mineral supplementation in Jersey steers. In Holstein steers, the levels of glutamine (No. 37) and alanine (No. 42) increased, whereas those of histidine (No. 50) and lysine (No. 52) decreased with increasing mineral supplementation.

In the case of Holstein steers, PLS–DA of 54 metabolites showed that all the treatment groups were separated from each other (Figure 4a). In contrast, Con and HM were separated from each other; however, NM had a tendency to separate from others in Jersey steers (Figure 4b). In PLS–DA, alanine, malic acid, proline, glycolic acid, histidine, palmitic acid, citric acid, leucine, isoleucine, γ-linolenic acid (γ-C_18:3_), pipecolic acid, succinic acid, 2-hydroxyglutaric acid, tyrosine, and glutamine showed top-15 ranked variable importance in the projection (VIP) scores for the evaluation of differentiation among the treatment groups in Holstein steers (Figure 5a). In Jersey steers, pyruvic acid, succinic acid, lactic acid, γ-linolenic acid (γ-C18:3), palmitoleic acid (C16:1), serine, myristic acid, histidine, leucine, α-linolenic acid (α-C18:3), palmitic acid, tyrosine, malic acid, oleic acid (C18:1), and 13-Methyltetradecanoic acid (C14:0) showed top-15 ranked variable importance in the projection (VIP) scores for evaluating the differentiation among treatment groups (Figure 5b). Among these, alanine was the most important serum metabolite in Holstein cattle, whereas pyruvic acid and succinic acid were the top-ranked metabolites with VIP scores > 2 in Jersey steers.

## 4. Discussion

HS causes several adverse effects on ruminants, and several strategies have been adopted to minimize these effects, such as improving the farm facilities and feed formulation, and dietary supplementation. This study focused on a higher concentration of dietary mineral supplementation to prevent the adverse effects of HS in ruminants. We considered both heat-sensitive and heat-tolerant breeds, Holstein and Jersey steers, respectively, to assess the mineral effects on breeds. Furthermore, the temperature humidity index (THI) was recorded during the feeding trial which was 82.79 ± 1.10, confirming the moderate HS condition of the experimental animals. This is in agreement with Armstrong [50] who categorized ambient conditions into five groups based on the THI: comfort zone (THI: <71), mild HS (THI: <72 to <79), moderate HS (THI: <80 to <90), severe HS (THI: >90), and death zone (THI: >100). Hahn et al. [51] also reported THI > 74 as HS in beef cattle, whereas Vitali et al. [52] stated that THI > 72 is believed to be HS for dairy cattle.

Blood biochemical parameters are important indicators of physiological changes in the animal body in response to adverse environmental conditions [53]. HS significantly decreased blood glucose (major metabolites of energy metabolism) and BUN concentrations in Holstein and Jersey steers, which was speculated to be due to reduced feed intake [54,55]. In contrast, increased BUN in sheep has been reported during HS due to reduced blood flow to the kidneys [56]. Joo et al. [54] further reported a significant reduction in serum protein, CHOL, and macrominerals such as Ca, Mg, and P levels in both Holstein and Jersey steers during heat stress. In the present study, all the aforementioned serum biochemical parameters were similar among the treatment groups in both Holstein and Jersey steers, indicating that dietary mineral concentrations do not affect serum biochemical parameters. This finding is supported by the non-significant DMI among different treatment groups in this study, and it is not unexpected because the present study was conducted during HS conditions with high THI (82.79 ± 1.10), and did not compare normal versus HS. In contrast, serum Cu and Zn concentrations increased, while Se concentration increased significantly by supplementing higher concentrations of these minerals, indicating their higher absorption in this study.

During HS, SOD activity must increase to prevent oxidative stress by neutralizing excess ROS in the host body [19]. Some minerals, especially Zn^2+^ and Cu^2+^, enhanced the SOD enzyme activity by acting as co-factors [23,24,57]. Se mainly acts as a cofactor for glutathione peroxidase [58,59]; however, increased SOD activity with Se supplementation has also been reported in broilers [60]. In contrast, HS causes oxidative stress through increased ROS production, which subsequently induces the activation of the HSP gene and the production of HSP [60,61,62]. Kim et al. [63] also reported that HS gradually increases HSPs in beef calves. Dietary Se supplementation lowered the expression of HSPs in broilers [58]. In the present study, significantly higher concentrations of SOD and lower concentrations of HSP27 and HSP70 were observed in the HM-supplemented groups compared to the Con group in both breeds, indicating their beneficial effect in preventing the adverse effects of HS, particularly oxidative stress, in both breeds. This might be due to the increased scavenging of ROS through SOD activity, which subsequently reduces the induction of the HSP gene. This finding is in agreement with that of Kumbhar et al. [60] who reported lower gene expression of HSPs along with higher SOD activity in broilers.

In the star patterns analysis, altered 54 metabolite levels were easily monitored in serum samples of all groups. The star patterns were characteristic and enabled visual recognition of the treated groups from the normal Control group. During HS, a negative energy balance develops owing to reduced feed intake, which leads to lower levels of whole-body glucose [64,65]. The liver plays a significant role in overcoming this situation through gluconeogenesis and by producing glucose from ruminal propionate, muscle tissue amino acids, and adipose tissue in the form of glycerol [66]. HS also alters serum metabolic pathways such as glycolysis, TCA cycles, beta-oxidation, and amino acid metabolism in cattle [67]. During carbohydrate metabolism, pyruvic acid is formed, which is further converted into acetyl-CoA and enters the TCA cycle through the aerobic pathway, whereas lactic acid is the metabolic end product of glucose metabolism in anaerobic pathways [68,69]. Tian et al. [12] reported that HS increases glycolysis, leading to higher amounts of pyruvic and lactic acid production to increase energy demand. The level of glycolysis during HS also varies among different cattle breeds [67]. In this study, the increasing pattern of pyruvic acid levels along with the higher concentration of mineral supplementation in Jersey steers might corroborate their metabolic adaptation to enhance energy supply during HS. In contrast, mineral supplementation did not affect the metabolic intermediates of glycolysis in Holstein steers, which might have caused breed-specific metabolic adaptation. HS upregulates the TCA cycle, leading to higher levels of metabolic intermediates, such as citric acid, α-ketoglutaric acid, succinic acid, fumaric acid, and malic acid [13,70]. In this study, decreasing patterns of succinic acid and malic acid, along with increased mineral supplementation, were observed in both breeds. This finding suggests that mineral supplementation at a higher dose prevents the HS-associated upregulation of the TCA cycle in both Holstein and Jersey steers. Hippuric acid is formed in the liver, enters the bloodstream, and is excreted through urine, and is suggested as a urinary biomarker of heat-stressed dairy animals [71]. The decreased pattern of the level of serum hippuric acid in both breeds, along with the increased mineral supplementation in this study, further strengthens the beneficial effect of additional mineral supplementation during HS. During HS, increased levels of circulating free fatty acids are produced through lipid catabolism [13]. Furthermore, HS increased acetyl-CoA formation from free fatty acids in the liver through β-oxidation to accelerate the TCA cycle [13,67]. In the present study, lower levels of γ-linolenic acid and 13-methyltetradecanoic acid were observed in the HM group compared to the other groups, and lower levels of linoleic acid and α-linolenic acid were found in both the HM and NM groups than in the Control group in both breeds. This finding suggests that higher levels of mineral supplementation prevented HS-associated lipid catabolism and β-oxidation in both steers. HS influences the production of higher levels of glucogenic amino acids, such as phenylalanine, tyrosine, methionine, alanine, glutamine, valine, and proline, which serve as gluconeogenesis precursors [67,72,73]. Likewise, in the present study, increased levels of phenylalanine in Jersey steers and alanine and glutamine in Holstein steers were observed with mineral supplementation, suggesting their contribution as gluconeogenesis precursors. However, decreased levels of tyrosine in both breeds, along with a higher concentration of mineral supplementation, suggested a greater conversion into glucose [74]. Some earlier studies indicated that ruminants preferentially use glucose as an energy source as it produces less metabolic heat [75,76]. In the PLS–DA analysis, different treatment groups were separated based on the VIP scores of the top 15 metabolites in both breeds, suggesting that all treatment groups in both breeds had their own metabolic adaptation pattern during HS; however, the metabolic pattern was mostly related to glucose homeostasis through gluconeogenesis.

## 5. Conclusions

Supplementation of dietary minerals high in organic selenium increased serum SOD concentrations, while decreasing HSP concentrations in both breeds. Star pattern recognition analysis revealed that the levels of succinic acid, malic acid, γ-linolenic acid, 13-methyltetradecanoic acid, and tyrosine decreased, whereas palmitoleic acid increased with increasing mineral concentrations in both breeds. In the PLS–DA analysis, based on the VIP scores of the top-15 ranked metabolites, different treatment groups were separated from each other in both breeds. Overall, the results suggested that dietary mineral supplementation in the HM group can be effective in preventing heat stress-associated oxidative stress and metabolic alterations in Holstein and Jersey steers. However, this study has some limitations, such as the low number of experimental animals and a short period of naturally adapted heat stress, which will be improved in future studies to reveal the detailed mechanisms of oxidative stress and their preventive measures using suitable antioxidants.

## Figures and Tables

**Figure 1 animals-12-03104-f001:**
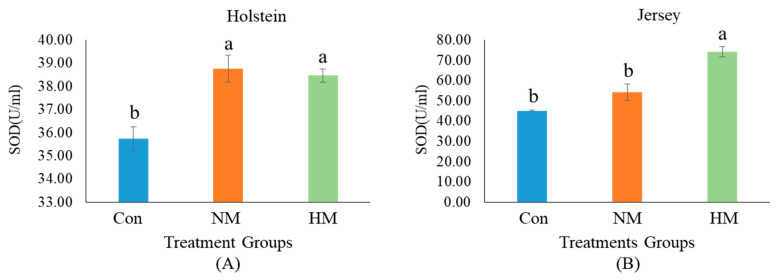
Serum superoxide dismutase (SOD) concentration of Holstein (**A**) and Jersey steers (**B**) supplemented with different concentrations of minerals. Con: only total mixed ration (TMR) (without mineral supplementation), NM: TMR + NRC recommended concentration of mineral supplementation (Se 0.1 ppm + Zn 30 ppm + Cu 10 ppm)/kg DM and HM: TMR + higher than recommended concentration of mineral supplementation (Se 3.5 ppm + Zn 350 ppm + Cu 28 ppm)/kg DM. The symbols a and b above the error bars in each figure indicate significant (*p* < 0.05) differences among treatment groups.

**Figure 2 animals-12-03104-f002:**
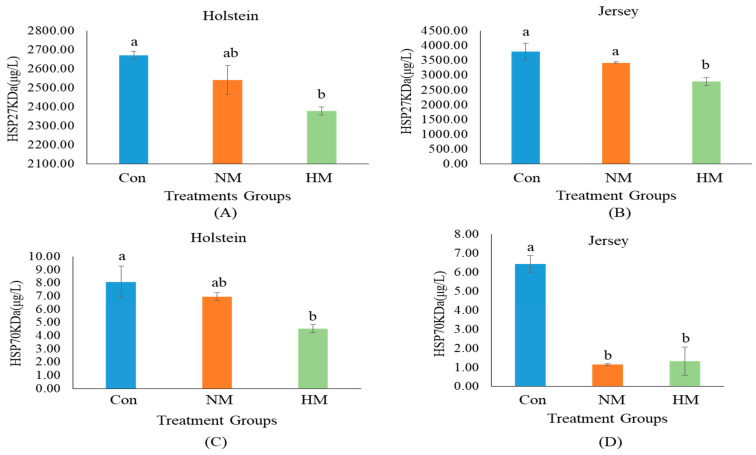
Serum concentrations of heat-shock proteins (HSPs) in Holstein (**A**,**C**) and Jersey steers (**B**,**D**) supplemented with different concentration of minerals. Con: only TMR (without mineral supplementation), NM: TMR + NRC recommended concentration of mineral supplementation (Se 0.1 ppm + Zn 30 ppm + Cu 10 ppm)/kg DM and HM: TMR + higher than recommended concentration of mineral supplementation (Se 3.5 ppm + Zn 350 ppm + Cu 28 ppm)/kg DM. The symbols a and b above the error bars in each figure indicate significant (*p <* 0.05) differences among treatment groups.

**Figure 3 animals-12-03104-f003:**
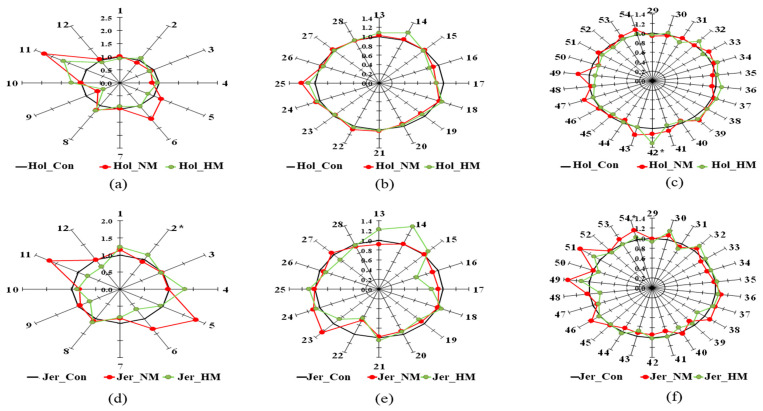
Star symbol plots of OAs (**a**,**d**), FAs (**b**,**e**), and AAs (**c**,**f**) in the sera of Holstein (**a**–**c**) and Jersey steers (**d**–**f**) supplemented with different concentrations of minerals. Hol, Holstein; Jer, Jersey; Con: only TMR (without mineral supplementation), NM: TMR + NRC recommended concentration of mineral supplementation (Se 0.1 ppm + Zn 30 ppm + Cu 10 ppm)/kg DM and HM: TMR + higher than recommended concentration of mineral supplementation (Se 3.5 ppm + Zn 350 ppm + Cu 28 ppm)/kg DM. Rays: 1 = Pyruvic acid, 2 = lactic acid, 3 = Glycolic acid, 4 = 2-Hydroxybutyric acid, 5 = 3-Hydroxypropionic acid, 6 = 3-Hydroxybutyric acid, 7 = Succinic acid, 8 = α-Ketoglutaric acid, 9 = Malic acid, 10 = 2-Hydroxyglutaric acid, 11 = Citric acid, 12 = Hippuric acid, 13 = Myristic acid (C14:0), 14 = Palmitoleic acid (C16:1), 15 = Palmitic acid (C16:0), 16 = γ-Linolenic acid (γ-C18:3), 17 = Linoleic acid (C18:2), 18 = Oleic acid (C18:1), 19 = α-Linolenic acid (α-C18:3), 20 = Stearic acid (C18:0), 21 = Arachidonic acid (C20:4), 22 = Eicosadienoic acid (C20:2), 23 = Arachidic acid (C20:0), 24 = Docosatetraenoic acid (C22:4), 25 = Docosapentaenoic acid (C22:5), 26 = 13-Methyltetradecanoic acid (C14:0), 27 = 14-Methylpentadecanoic acid (C15:0), 28 = 16-Methylheptadecanoic acid (C17:0), 29 = Tryptophan, 30 = Phenylalanine, 31 = Tyrosine, 32 = Leucine, 33 = Methionine, 34 = Isoleucine, 35 = Valine, 36 = Pipecolic acid, 37 = Glutamic acid, 38 = α-Aminobutyric acid, 39 = Proline, 40 = Hydroxyproline,, 41 = Threonine, 42 = Alanine, 43 = Serine, 44 = Creatine, 45 = Glutamine, 46 = Creatinine, 47 = Asparagine, 48 = Citrulline, 49 = 1-Methylhistidine, 50 = Histidine, 51 = 3-Methylhistidine, 52 = Lysine, 53 = Ornithine, 54 = Arginine.

**Figure 4 animals-12-03104-f004:**
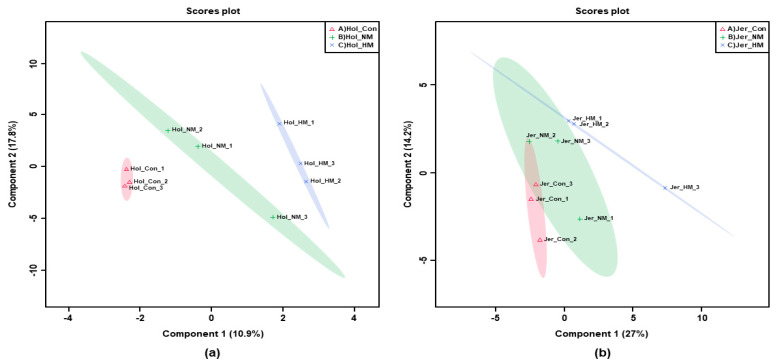
PLS−DA analyses for the serum metabolites of Holstein (**a**) and Jersey (**b**) steers supplemented with different concentration of minerals. Con: only TMR (without mineral supplementation), NM: TMR + NRC recommended concentration of mineral supplementation (Se 0.1 ppm + Zn 30 ppm + Cu 10 ppm)/kg DM and HM: TMR + higher than recommended concentration of mineral supplementation (Se 3.5 ppm + Zn 350 ppm + Cu 28 ppm)/kg DM. The numbers 1, 2, and 3 inside the figure indicate the number of animals.

**Figure 5 animals-12-03104-f005:**
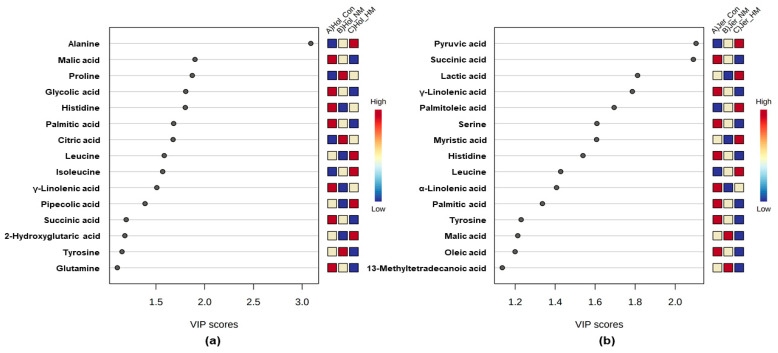
Variable importance analysis of top 15 serum metabolites in Holstein (**a**) and Jersey (**b**) steers supplemented with different concentration of minerals. Hol: Holstein, Jer: Jersey, Con: only TMR (without mineral supplementation), NM: TMR + NRC recommended concentration of mineral supplementation (Se 0.1 ppm + Zn 30 ppm + Cu 10 ppm)/kg DM and HM: TMR + higher than recommended concentration of mineral supplementation (Se 3.5 ppm + Zn 350 ppm + Cu 28 ppm)/kg DM.

**Table 1 animals-12-03104-t001:** Chemical composition of total mixed ration (TMR).

Ingredients	Compositions (% of DM)
Corn gluten feed	8.40
Soybean	6.24
Beet pulp	4.20
Wheat bran	3.15
Corn flakes	2.21
Molasses	1.04
Rice wine residue	5.25
Brewer’s grain residue	21.01
Annual ryegrass straw	27.29
Orchard grass straw	21.01
Limestone	0.10
Sodium bicarbonate	0.01
Salt	0.09
Total	100.00
**Chemical composition (DM basis)**	**% or ppm**
DM (fresh basis)	58.98%
CP	13.55%
Crude Fiber	21.92%
Crude fat	3.02%
Ash	9.21%
Calcium	1.22%
Phosphorus	0.47%
NDF	48.00%
ADF	25.36%
Zinc	77.35 ppm
Copper	17.31 ppm
Selenium	0.05 ppm

DM, dry matter; CP, crude protein; NDF, neutral detergent fiber; ADF, acid detergent fiber.

**Table 2 animals-12-03104-t002:** Temperature–humidity index (Mean ± SD) of the experimental periods.

Period	Mean Ambient Temp (°C)	rH (%)	THI
Period 1	30.50 ± 0.60	80.67 ± 5.63	83.77 ± 0.55
Period 2	28.49 ± 1.24	88.13 ± 3.98	81.60 ± 2.01
Period 3	29.74 ± 1.35	83.86 ± 6.15	83.00 ± 1.69
Average	29.58 ± 1.02	84.22 ± 3.74	82.79 ± 1.10

rH, relative humidity; THI, temperature-humidity index.

**Table 3 animals-12-03104-t003:** Growth performance, serum biochemistry, trace mineral, superoxide dismutase (SOD), and heat-shock proteins (HSPs) concentrations of Holstein and Jersey steers supplemented with different concentration of minerals.

Parameters	Breed	Treatment			SEM ^(4)^		GLM *p*-Value ^(5)^	Mixed *p*-Value ^(6)^		
		Con ^(1)^	NM ^(2)^	HM ^(3)^	IN	All		B	T	B × T
IBW	Holstein	783.67	782.33	782.00	25.355	19.174	0.9988	<0.0001	0.9934	0.9995
	Jersey	579.67	577.00	577.00	12.994		0.9895			
DMI (kg/d)	Holstein	12.74	13.32	12.90	1.541	1.172	0.9665	0.1250	0.9474	0.9816
	Jersey	11.02	11.29	11.37	0.803		0.9571			
FBW	Holstein	793.00	792.33	792.33	25.265	19.246	0.9998	<0.0001	0.9992	0.9974
	Jersey	592.67	595.00	594.00	13.227		0.9941			
ADG (kg)	Holstein	0.78	0.83	0.86	0.090	0.096	0.877	0.0002	0.1476	0.3498
	Jersey	1.08	1.50	1.42	0.102		0.1250			
FE	Holstein	0.06	0.07	0.07	0.009	0.010	0.8830	<0.001	0.2305	0.4543
	Jersey	0.10	0.13	0.13	0.012		0.2427			
Ca (mg/dL)	Holstein	8.97	8.80	8.73	0.179	0.118	0.6643	0.0482	0.0931	0.8349
	Jersey	9.17	9.00	8.90	0.075		0.2519			
Mg (mg/dL)	Holstein	2.28	2.44	2.41	0.079	0.070	0.3743	0.1064	0.5210	0.0625
	Jersey	2.62	2.39	2.57	0.071		0.1394			
P (mg/dL)	Holstein	6.95	7.15	7.30	0.367	0.392	0.8465	0.7570	0.7987	0.9852
	Jersey	7.15	7.20	7.40	0.417		0.9235			
BUN (mg/dL)	Holstein	10.50	9.00	9.00	1.167	0.778	0.7542	0.5529	0.2266	0.9033
	Jersey	11.50	9.67	9.00	0.389		0.1003			
Glu (mg/dL)	Holstein	81.33	78.50	78.50	0.889	2.170	0.1280	0.2234	0.5971	0.9513
	Jersey	78.50	76.33	76.00	3.229		0.8587			
TP (g/dL)	Holstein	8.05	8.30	8.75	0.300	4.103	0.5801	0.2822	0.3284	0.3151
	Jersey	8.45	8.20	8.05	0.238		0.5827			
CHOL (mg/dL)	Holstein	138.00	139.50	128.50	7.333	11.540	0.6413	0.6214	0.9427	0.7271
	Jersey	148.50	141.33	141.00	12.734		0.9018			
Cu (μg/L)	Holstein	740	830	940	50.00	100.34	0.1542	0.3943	0.7605	0.8748
	Jersey	880	970	1080	50.00		0.2162			
Zn (μg/L)	Holstein	750	855	910	71.67	50.00	0.4562	0.0235	0.0346	1.000
	Jersey	815	900	930	71.44		0.6261			
Se (μg/L)	Holstein	106.50 ^c^	142.00 ^b^	218.50 ^a^	4.000	8.825	0.0017	0.0626	0.0004	0.2701
	Jersey	77.00 ^b^	143.33 ^a^	174.50 ^a^	13.651		0.0383			

^(1)^ Con: only fed basal total mixed ration (TMR) (without additional mineral supplementation). ^(2)^ NM: fed TMR and NRC recommended concentration of mineral supplementation (Se 0.1 ppm + Zn 30 ppm + Cu 10 ppm) as DM basis. ^(3)^ HM: fed TMR and higher than recommended concentration of mineral supplementation (Se 3.5 ppm + Zn 350 ppm + Cu 28 ppm) as DM basis. ^(4)^ Standard error of the means of different treatment (trace mineral supplementation) groups in individual (IN) breed and all. ^(5)^
*p* value received from different treatment (trace mineral supplementation) groups in each breed by General linear model. ^(6)^
*p* value received from the Mixed procedure of SAS with breed effect (B), treatment effect (T; trace mineral supplementation), and the interaction effect between breed and treatment (B × T). IBW, initial body weight; DMI, dry matter intake; Ca, calcium; FBW, final body weight; ADG, average daily gain; FE, feed efficiency; Mg, magnesium; P, phosphorus; BUN, blood urea nitrogen; Glu, glucose; TP, total protein; AST, aspartate aminotransferase; CHOL, cholesterol; Cu, copper; Zn, zinc; Se, selenium. ^a,b,c^ in the same row indicate the significant differences (*p* < 0.05) of data among three different treatments regardless of breed.

## Data Availability

Data are available upon request to the corresponding author.

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
