# Peer review of "Higher Concentration of Dietary Selenium, Zinc, and Copper Complex Reduces Heat Stress-Associated Oxidative Stress and Metabolic Alteration in the Blood of Holstein and Jersey Steers"

_animals, 2022, doi:10.3390/ani12223104_

Round 1

Reviewer 1 Report

Please see the attached file for comments. 

Author Response

Reviewer 1:

Please see the attached file for comments. 

Comments to authors

The article under review comprehensively presented the blood biochemistry in response to the mineral supplementation during heat stress. The study is very important considering the heat stress challenges for livestock due to climate change. The authors have already done good job in writing the manuscript. A couple of comments are given below to further strengthen the quality of the manuscript.

Response: Thank you for your kind review and nice comments. It is noted that all the corrections and modifications of this revised manuscript were made by using the Track Change option.

Abstract:

The abstract was well written. If possible, please add the findings related to glucose homeostasis in the abstract. Please see the comment on Line 488-490.

Response: As per the reviewer's comment, we added information related to glucose homeostasis in the abstract section (Lines 45-47).

Line 36: Would it be “tended to have higher alanine levels” or “tentatively higher alanine levels”.

Response: “tended to have higher alanine levels” (Line 42).

Line 42: The keyword “Holstein and Jersey steers” could be replaced with “ruminants”. It would help other researcher, working on heat stress, to find this study.

Response: We modified it as per the reviewer's suggestion (Line 52).

Introduction:

The authors constructed strong arguments to establish the need of the study. It would be great if the authors remove the text from line 77-81 and place it after the first sentence of the introduction section. It will increase the flow of the text about heat stress challenges.

Response: We modified it as per the reviewer's suggestion (Lines 56-60).

Materials and Methods:

The authors sufficiently explained the details of materials used and methods applied. Great job. The heat stress in ruminants is cyclic and some physiological responses (respiration rare and body temperature) are incidental, high at peak heat stress and low at early morning hours. Is it the same with blood biochemistry? Would it be possible that if the blood sampling was done at peak hours of the heat stress like afternoon or after sunset, some responses would have been different? Please comments on it.

Response: Actually, we conducted this experiment in naturally adapted cattle under heat stress conditions during summer and the temperature-humidity index (THI) was considered the key indicator of heat stress in this study. We collected blood samples one time before the morning meal. We will definitely consider your valuable and interesting suggestions regarding sample collection for our future studies.   

Results and Discussion:

The results were well presented, and the discussion was fully justified. a small suggestion is given below.

Line 488-490: The glucose metabolism in ruminants during heat stress is an interesting area to be further explored. Some studies indicated that ruminants preferentially use glucose as an energy source as it produces less metabolic heat. (Baumgard and Rhoads, 2007; 2013). The glucose metabolism is very important in lactating cows as it is the precursor of lactose that determines the quantity of milk. The finding of current study, indicating that the metabolic pattern was mostly related to glucose homeostasis through gluconeogenesis, strengthens the previous research. It should be discussed and be added in the abstract. This would be used as a reference for future studies on heat stress in ruminants because it is much easier to determine the blood glucose levels than the whole array of related metabolites.

Baumgard L.H., and Rhoads R.P., 2007. The effects of hyperthermia on nutrient partitioning. In Proceedings of Cornell Nutritional Conference for Feed Manufacturers. Cornell University, New York, USA. p 93-104.

Baumgard, L.H. and Rhoads Jr, R.P., 2013. Effects of heat stress on postabsorptive metabolism and energetics. Annual Review of Animal Biosciences, 1(1), 311-337.

Response: We updated the discussion regarding glucose homeostasis as per the reviewer's recommendation (Lines 540-541).

Reviewer 2 Report

In this study, the author investigated the effect of recommended and high mineral concentrations on antioxidant enzymes, heat shock proteins, and metabolites in the blood during heat stress of Holstein and Jersey steers. And evaluated the changes between heat stress-related oxidative stress and metabolites in the blood of animals supplemented with high concentrations of minerals. It was ultimately concluded that increased mineral supplementation during periods of high temperature is an effective strategy for reducing heat stress. However, in the manuscript, there are a lot of irregular basic writing formats, no use of international standard units, and grammatical confusion. Please correct them one by one.

1.      Line 30-32, please clarify the writing form of experimental grouping.

2.      Line 33-35, Please standardize the writing format.

3.      In this article, there are large amounts of ppm, ppm/kg, ng/ml, μg/mL, and so on. Units and formats are not uniform, please write in SI units.

4.      Line 216, P in P<0.05 should be in italics, please check the entire article for consistent formatting.

5.      In Figure 1 and 2, the “a, b” on the error bar and “figure (a), and (b)” should be represented by different symbols.

6.      The conclusion is not simply a repetition of the results section. Instead, it should explain what problems the results show, what the research value and significance are, what problems still exist in the research process, and how to solve them. The author should rewrite the conclusion section.

7.      The grammar of this manuscript should be revised.

8.    For innovative articles, references should be nearly 3-5 years. Please update references.

Author Response

Reviewer 2:

In this study, the author investigated the effect of recommended and high mineral concentrations on antioxidant enzymes, heat shock proteins, and metabolites in the blood during heat stress of Holstein and Jersey steers. And evaluated the changes between heat stress-related oxidative stress and metabolites in the blood of animals supplemented with high concentrations of minerals. It was ultimately concluded that increased mineral supplementation during periods of high temperature is an effective strategy for reducing heat stress. However, in the manuscript, there are a lot of irregular basic writing formats, no use of international standard units, and grammatical confusion. Please correct them one by one.

Response: Thank you for your kind and keen review. It is noted that all the corrections and modifications of this revised manuscript were made by using the Track Change option.

  1. Line 30-32, please clarify the writing form of experimental grouping.

Response: As per the reviewer's recommendation, we clarified the experimental grouping (Lines 33-37).

  1. Line 33-35, please standardize the writing format.

Response: As per the reviewer's suggestion, we standardized the writing format (Lines 38-41).

  1. In this article, there are large amounts of ppm, ppm/kg, ng/ml, μg/mL, and so on. Units and formats are not uniform, please write in SI units.

Response: For serum parameters, μg/L is used as SI units. Therefore, we modified some of the parameters as μg/L where relevant; however, values of some other parameters were low and difficult to understand if we use the μg/L. In those instances, we did not change. In addition, ppm is a widely used measurement unit for mineral concentration in the feed ingredients and we preserved it for dietary mineral supplementation.

Just for your kind references:

1 ng/ml = 1 μg/L 

1 μg/dL = 10 μg/L

1 ng/μl = 1000 μg/L 

1 mg/dl = 10000 μg/L

1 g/dl = 10000000 μg/L

1 ppm = 1000 μg/L

  1. Line 216, P in P<0.05 should be in italics, please check the entire article for consistent formatting.

Response: We revised it as per the reviewer's recommendation (Line 244).

  1. In Figure 1 and 2, the “a, b” on the error bar and “figure (a), and (b)” should be represented by different symbols.

Response: As per the reviewer's suggestion, we modified Figures 1 and 2.

  1. The conclusion is not simply a repetition of the results section. Instead, it should explain what problems the results show, what the research value and significance are, what problems still exist in the research process, and how to solve them. The author should rewrite the conclusion section.

Response: As per the reviewer's recommendation, we revised the conclusion section (Lines 547-563).

  1. The grammar of this manuscript should be revised.

Response: As per the reviewer's recommendation, we revised the grammar of this manuscript. We will perform English editing one more if necessary in the final stage of production.

  1.   For innovative articles, references should be nearly 3-5 years. Please update references.

Response: As per the reviewer's recommendation, we updated it by adding some recent relevant references (nearly 3-5 years).

Reviewer 3 Report

The manuscript entitled "Higher concentration of dietary selenium, zinc, and copper complex reduces heat stress-associated oxidative stress and metabolic alteration in the blood of Holstein and Jersey steers" has been reviewed. The study is unrealistic, because higher dietary selenium, zinc, and copper would finally lead to polution (soil, water, etc.). On the other hand, more practical methods could be explored to relieve the heat stress, e.g., diet composition, surroundings. Another major concern would come to the experimental design itself, where 3×3 Latin square design were taken and a total of 3 cattle, which means actually n=3. This make the current data hard to believe, and many details have been missed, e.g., Statistical analysis, which is fixed factor, random factor.

Author Response

Reviewer 3:

The manuscript entitled "Higher concentration of dietary selenium, zinc, and copper complex reduces heat stress-associated oxidative stress and metabolic alteration in the blood of Holstein and Jersey steers" has been reviewed. The study is unrealistic, because higher dietary selenium, zinc, and copper would finally lead to polution (soil, water, etc.). On the other hand, more practical methods could be explored to relieve the heat stress, e.g., diet composition, surroundings. Another major concern would come to the experimental design itself, where 3×3 Latin square design were taken and a total of 3 cattle, which means actually n=3. This make the current data hard to believe, and many details have been missed, e.g., Statistical analysis, which is fixed factor, random factor.

Responses:

-Thank you for your kind review and comments.

-We partially agree with the following statement- ‘The study is unrealistic, because higher dietary selenium, zinc, and copper would finally lead to pollution (soil, water, etc.).’ However, we did not exceed the maximum tolerable level of those minerals recommended by NRC which will pose a negative effect on animals and the environment. Here, we focused to reduce oxidative stress by dietary supplementation of antioxidants at higher concentrations. We already considered some other approaches to reduce the adverse effect of heat stress mentioned by the reviewer in our ongoing study.

-We agree that the more animals the less error in the experimental data. We used two different breeds (3 Cattle in each group) to minimize the error to a certain extent. We already increased the number of animals in our ongoing study and will be considered in subsequent studies in future.

Reviewer 4 Report

describe how the animals not included in the study were used before the study was started

Author Response

Reviewer 4:

describe how the animals not included in the study were used before the study was started

Responses:

-Thank you for your kind review.

-We selected those animals that were reared on the farm under normal feeding conditions with good health and similar body weight for each breed. Also, we did not use those animals for any other experiments for the last 6 months before the study started.

Reviewer 5 Report

This manuscript describes an experiment evaluating supplementation of minerals with roles in oxidative stress on the effects of heat stress in dairy steers. Overall, the study was conducted well and provides meaningful results. I do have a few major concerns:1) why was Mn not included due to its role in Mn-SOD, 2) animal performance and THI data are not shown, 3) why was serum instead of liver mineral levels measured, and 4) results and discussion of the metabolomics data are difficult to follow and I don't think authors interpretation is supported by the data in some instances.

Specific comments:

L77-84 - this section on  climate change and breed should be moved to the first paragraph of the introduction

L93 - it seems there could be carryover effects on mineral levels in a Latin square design. how did the authors ensure the mineral status of animals returned to baseline between periods?

L95-97 - it is unclear if these mineral concentrations are the concentration of the mineral supplement fed (if so then the amount of supplement needs to be reported) or if they are the concentration of the final diet

L112 - ADG was not measured, it was calculated from measured body weight. how was initial and final BW measured?

Table 1 - Zn and Cu levels in the basal TMR exceed the NRC requirements, but Se does not. How did/could this effect results?

L126 - why not take liver biopsy instead of serum? liver is a better measure of mineral status.

L175 - diethyl ether is mentioned twice

L202-211 - why were composition rather than concentration values used for data analysis? What is the purpose of normalizing values to the control? The Kruskal-Wallis test needs to be described in this section.

L214 - The statistical analysis does not seem correct for a Latin square. the factors used in the model need to be described and based on Table 2, animal/row and period/column are not included in the model. also, it may be better to describe the experimental design as a replicated latin square with breed as the replicate.

L218 - Temperature, humidity, and THI data among treatment should be presented unless all animals were housed in the same building in which case a graph with these data over time would be useful

L218 - a Table 2 should be added with animal performance (BW, ADG, DMI, gain:feed) data

L228 - This sentence sounds like it is referring to the treatment effect which has a p-value of 0.14, but the interaction effect p-value is reported

L229 - change this sentence to clarify that these are the results for the interaction effect

Table 2 - it is concerning that treatment did not affect mineral concentrations. in the footnote, please clarify that the concentrations mentioned are mineral supplement or total diet, and adjust for all figures and tables please. I do not understand the p-value columns for holstein and jersey because your statistical model included breed as a factor

Figure 1 and 2 - since SOD and HSP had significant interaction, I agree that it is better to present in graphical form, but then the data does not need to be presented in Table 2 and the p-values need to be included in captions of figures. either way superscripts need to be explained in captions

L303-321 - in star pattern recognition analysis how do you determine what is significantly different - I do not see any statistics presented and the description of results here does not match the Kruskal-Wallis test in Tables 3 and 4. I also disagree that some of the metabolites described as different among treatments are different - for example No. 7 for holstein steers.

L324-325 - HM is mentioned twice in this sentence

L337 - I think you need to finish the sentence with 'in Jersey steers.'

Table 3 and 4 - is the VIP score computed within groups (OA, FA, AA) or across all groups of metabolites? Since you show the important data in Figures 3-5, these tables could be supplemental tables

Figure 4 - what does the 1, 2, and 3 number for each marker mean? I think it is each of the 3 animals, but that needs to be explained in the figure caption

Figure 5 - change caption to 'serum metabolites in Holstein'

L441 - the increase in SOD and reduction in HSP indicates better response to heat stress, but the lack of response in DMI (L425) is interesting. If oxidative stress is relieved, then what is driving the decrease in DMI? If relieving oxidative stress does not improve DMI and animal performance then what is the benefit of feeding the additional minerals?

L447-490 - based on comments above for interpretation of star pattern recognition, this discussion may need to be revised

L491 - the strongest evidence is the increase in SOD and decrease in HSP, but the changes in metabolites from glycolysis and gluconeogenesis is weak evidence based on interpretation of Kruskal-Wallis test.

Author Response

Dear Reviewer 5,

Please see the responses in the attached file.

Thank you.

Round 2

Reviewer 2 Report

The comments of answer is very good.

Author Response

Thank you very much for your kind review.

Reviewer 3 Report

The revised version seems to answer another questions, not my main concerns and suggestions. Another concerns would be the language, please find a native English speaker to polish the manuscript. 

Author Response

(The authors gave the same response as above.)

Reviewer 5 Report

The authors have improved this manuscript. I still have some major concerns 1) 7 day washout period seems too short for whole body mineral levels to return to baseline - do you have any data to support return to baseline, 2) statistical analysis still does not account for animal and period in a Latin square analysis, and 3) Se was the only deficient mineral and so likely the response to mineral supplementation is solely due to Se, but this cannot be discerned.

Specific comments:

L117 - is 7 days really long enough for whole body mineral concentrations to return to baseline. I have doubts.

L128 - I do not see the description of measuring IBW, and FBW with the calculation of ADG.

Table 1 - It still seems problematic that Zn and Cu are above requirements in the basal TMR and serum levels did not significantly increase with supplementation (this could be the dynamic nature of serum). Based on the data, the conclusion is that the treatment effect is due to Se

L238 - there is no description of how animal and period were handled in the statistical analysis. please add

L260 - describe the interaction

Table 3 - values for FE in Jersey steers seem incorrect

L536-541 - this section goes from talking about glucose, then explains what star pattern is, then talks about glucose again. I think the sentences on star pattern should be placed somewhere else.

Author Response

Author’s responses to the reviewer (2nd round)

Reviewer 5:

Comments and Suggestions for Authors

The authors have improved this manuscript. I still have some major concerns 1) 7 day washout period seems too short for whole body mineral levels to return to baseline - do you have any data to support return to baseline, 2) statistical analysis still does not account for animal and period in a Latin square analysis, and 3) Se was the only deficient mineral and so likely the response to mineral supplementation is solely due to Se, but this cannot be discerned.

Response: Thank you for your kind and keen review. It is noted that all the corrections and modifications of this revised manuscript were made by using the Track Change option. We have responded to all the issues raised by the respected reviewer in the specific comment section.

Specific comments:

L117 - is 7 days really long enough for whole body mineral concentrations to return to baseline. I have doubts.

Response: We agree that a long washing period is better to return the mineral concentration to baseline; however, we mentioned previously that this study was conducted using animals under naturally adapted heat stress conditions and the summer month in South Korea stands only for about 2-3 months. We will consider this issue with care in our future study and use an environmentally controlled house to conduct heat stress-related experiments. Moreover, we added some references here where they maintained a 5 to 7 days washing period to support our 7 days washing period in this study (see references 32-34).

Matamoros C, Salfer IJ, Bartell PA, Harvatine KJ. Effect of the timing of sodium acetate infusion on the daily rhythms of milk synthesis and plasma metabolites and hormones in Holstein cows. Journal of dairy science. 2022 Sep 1;105(9):7432-45.

Zeyner A, Romanowski K, Vernunft A, Harris P, Müller AM, Wolf C, Kienzle E. Effects of different oral doses of sodium chloride on the basal acid-base and mineral status of exercising horses fed low amounts of hay. Plos one. 2017 Jan 3;12(1):e0168325.

Fowler AL, Brümmer-Holder M, Dawson KA. Dietary trace mineral level and source affect fecal bacterial mineral incorporation and mineral leaching potential of equine feces. Sustainability. 2019 Dec 11;11(24):7107.

L128 - I do not see the description of measuring IBW, and FBW with the calculation of ADG.

Response: As per the reviewer’s suggestion, we added the calculation of ADG (Lines 122-124).

Table 1 - It still seems problematic that Zn and Cu are above requirements in the basal TMR and serum levels did not significantly increase with supplementation (this could be the dynamic nature of serum). Based on the data, the conclusion is that the treatment effect is due to Se

Response: Though the serum level of Cu and Zn were not increased significantly based on GLM analysis; however Zn level increased significantly among treatment groups regardless of breeds according to the mixed analysis. In contrast, Se showed a significant effect in both analyses. Therefore, we concluded the effect of Se as per the reviewer’s recommendation (Lines 530-531).

L238 - there is no description of how animal and period were handled in the statistical analysis. please add

Response: The model included the fixed effect of treatment and the random effects of animal and period where each animal was distributed under 3 different treatment groups in 3 different periods (Lines 228-230).

Con

NM

HM

Period 1

Animal 1

Animal 2

Animal 3

Period 2

Animal 2

Animal 3

Animal 1

Period 3

Animal 3

Animal 1

Animal 2

L260 - describe the interaction

Response:  We re-checked all the data presented in this manuscript in order to avoid any sort of unwanted mistake and find a few mistakes which have been corrected in this revised manuscript (Table 2). We revised the text as follows-

In contrast, serum concentrations of Cu and Zn were observed similar among the different treatment groups in both breeds (p>0.05). In addition, according to the mixed analysis, a numerical increase in the serum Cu concentration (p>0.05) while a significant increase (p<0.05) in the concentrations of serum Zn and Se were observed along with the supplementation of trace minerals regardless of breeds. However, Jersey steers had a significantly higher serum Zn concentration compared to Holstein steers regardless of treatment (p<0.05). (Lines 246-252).

Table 3 - values for FE in Jersey steers seem incorrect

Response: We confirmed and corrected FE data in Table 2.

L536-541 - this section goes from talking about glucose, then explains what star pattern is, then talks about glucose again. I think the sentences on star pattern should be placed somewhere else.

Response: As per the reviewer’s suggestion, we revised it (Line 478-480).